# Rainfall Variability and Rice Sustainability: An Evaluation Study of Two Distinct Rice-Growing Ecosystems

**Masoud K. Barati [1], V. S. Manivasagam [2,*] , Mohammad Reza Nikoo [3] , Pasoubady Saravanane [4] , Alagappan Narayanan [4] and Sudheesh Manalil [2,5]**

[1]  Amrita School for Sustainable Development, Amrita Vishwa Vidyapeetham, Amritapuri, Clappana PO, Kollam 690525, India

[2]  Amrita School of Agricultural Sciences, Amrita Vishwa Vidyapeetham, J. P. Nagar, Arasampalayam, Myleripalayam, Coimbatore 642109, India

[3]  Department of Civil and Architectural Engineering, Sultan Qaboos University, Muscat 123, Oman

[4]  Department of Agronomy, Pandit Jawaharlal Nehru College of Agriculture & Research Institute, Karaikal 609603, India

[5]  UWA Institute of Agriculture, UWA School of Agriculture and Environment, The University of Western Australia, Perth 6009, Australia

*   Correspondence: vs_manivasagam@cb.amrita.edu

**Abstract:** The inconsistency of the Indian monsoon has constantly threatened the country's food production, especially key food crops such as rice. Crop planning measures based on rainfall patterns during the rice-growing season can significantly improve the sustainable water usage for water-intensive crops such as rice. This study examines the variability of Indian monsoonal rainfall in rainfed and irrigated rice-cultivating regions to improve rainfall utilization and irrigation water-saving practices. Two distinct rice-growing conditions in southern peninsular India are chosen for this study. The preliminary seasonal rainfall analysis (1951–2015) showed anomalies in the Sadivayal (rainfed rice) region compared to the Karaikal (irrigated rice). The dry-spell analysis and weekly rainfall classification suggested shifting the sowing date to earlier weeks for the Thaladi season (September–February) and Kar season (May–September) to avoid exposure to water stress in Sadivayal. Harvesting of excess rainwater during the wet weeks is proposed as a mitigation strategy for Karaikal during the vegetative stage of the Kuruvai season (June–October) and Late Thaladi season (October–February), where deficit rainfall is expected. Results showed that an adaptation strategy of early sowing is the most sustainable measure for rainfed rice cultivation. However, harvesting the excess rainwater is an ideal strategy to prevent water stress during deficient rainfall periods in irrigated rice farming. This comparative study proposes a comprehensive rainfall analysis framework to develop sustainable water-efficient rice cultivation practices for the changing rainfall patterns.

**Keywords:** rainfall variability; sustainable cropping strategies; irrigation planning; rainfed rice; Cauvery delta; rainfall trend analysis

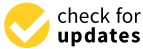



## 1. Introduction

The uneven distribution of the Indian monsoon often causes uncontrolled runoff and water scarcity issues, which negatively affect crop growth and water productivity, especially for major food crops such as rice and wheat [1,2]. Precipitation anomalies decreased water availability and affected crop production in India [3–7]. To resolve this issue, a detailed assessment of oscillations in rainfall patterns over a specific growing season is crucial for the field as well as regional-level water resource management and irrigation planning measures [8,9]. Previous studies highlighted supplemental irrigation (SI) and shifts in the sowing date as mitigation and adaptation strategies to prevent the pernicious effects of extreme climatic events. [10–14]. Hence, continuous monitoring and analysis of the

rainfall distribution, intensity, and monsoon onset and withdrawal dates are prerequisites to improve water management and planning on both local and regional scales.

Assessment of rainfall variability during the crop growth period would provide better insight into water availability and additional irrigation requirement for ideal crop cultivation. The Mann–Kendall test is one of the most common nonparametric methods used to analyze the rainfall trend [15–17]. To give an example, Duhan and Pandey [18] studied the spatiotemporal variations of 102-year rainfall in the Madhya Pradesh region of India. Pandey et al. [19] simulated and compared the influence of rainfall contribution in irrigated and rainfed farming. Rainfall scenario analysis indicated that the harvested rainwater helps to mitigate the effects of the dry period in the Indo–Gangetic Plain of India. Hasan et al. [20] determined that about 38% of the total consumptive water is used from rainwater harvesting for rice cultivation in Bangladesh. Thus, critical analysis of rainfall distribution and effective planning of harvesting excess water can be significantly used for crop cultivation.

Rice is the staple food of two-thirds of the world's population, including India, feeding almost three billion people [21]. It is highly vulnerable to climate change abnormalities [18,22]. Uneven distribution of rainfall, erratic onset and withdrawal of monsoon, and frequent interseason dry-spell occurrence hamper the rice yield. Spatial and temporal distribution analysis of rainfall during crop growth has influenced crop yield and water productivity in peninsular India. Farmers' cropping strategies and irrigation planning accounting for the conjunctive use of rainfall and irrigation have positively influenced crop yield [12,23]. Research studies on the influence of rainfall on rice cultivation in the Cauvery delta zone have so far been limited [24]. The irrigation system in the Cauvery delta zone includes floodwater storage facilities such as tanks on which rice water demand is heavily dependent. Irrigated rice farming in the deltaic region is indirectly dependent on the distribution of rainfall in the upper catchment of the Cauvery basin. Overexploitation of reservoir water reduces the water use efficiency in the field in due course, rice yield as well.

Since rice has been predominately cultivated as an irrigated crop in India, the previous studies have not given the focus on the rainfall variability and appropriate sustainable measures planning against the untenable rice cultivation. In the recent past, the decline of the traditional tank irrigation systems and growing competition for river water emphasize the importance of rainfall assistance to irrigation at a local scale [25]. In the agriculture sustainability domain, most of the global studies focused on soil erosion, land restoration, and rehabilitation [26–29]. To the best of our knowledge, there is no systematic framework for analyzing rainfall characteristics during the crop growth period for both rainfed and irrigated rice systems. The overarching aim of this study is (i) to analyze the rainfall variability for two distinct rice-cultivating regions in southern peninsular India and (ii) to assess the sustainable crop planning measures to prevent crop exposure to water shortages during various growth phases for rice crop. This study's findings can help plan sustainable rice cultivation measures with the effective use of monsoon rain.

## 2. Materials and Methods

### 2.1. Study Area

Sadivayal is a remote tribal village located on the foothills of Western Ghats, Coimbatore. Sadivayal farmers grow primarily rainfed rice once a year without irrigation facilities [30]. Accordingly, Sadivayal is taken as a representative region for rainfed rice cultivation. The deltaic region of the Cauvery River basin is one of the significant irrigated rice cultivating areas in peninsular India (Figure 1). Water availability at Mettur Reservoir is the lifeline for rice cultivation in the Cauvery Delta zone. In addition to the reservoir water, groundwater is widely used as a supplementary source of irrigation in this region for intensive agricultural production. Conversely, the pressure on the freshwater sources has increased due to erratic rainfall and decreasing river flow volume in the delta region [31]. Thus, the Karaikal region from the Cauvery delta zone is chosen as a representative region

for irrigated rice cultivation. The daily rainfall data covering the period 1951–2015 for both the locations are collected from the India Meteorological Department (for the Sadivayal region) and Pandit Jawaharlal Nehru College of Agriculture and Research Institute (for the Karaikal region). The major rice-growing seasons in Tamil Nadu are Kar, Kuruvai, Samba, Thaladi, Late Thaladi, and Navarai, as listed in Table 1. Kar and Kurvavai seasons coincide with the Southwest Monsoon (SWM). Similarly, Thaladi and Late Thaladi coincide with Northeast Monsoon (NEM). The length of rice growth duration is classified into three categories as (i) short duration: 100–120 days, (ii) medium duration: 121–140 days, and (iii) long duration: 141–160 days.

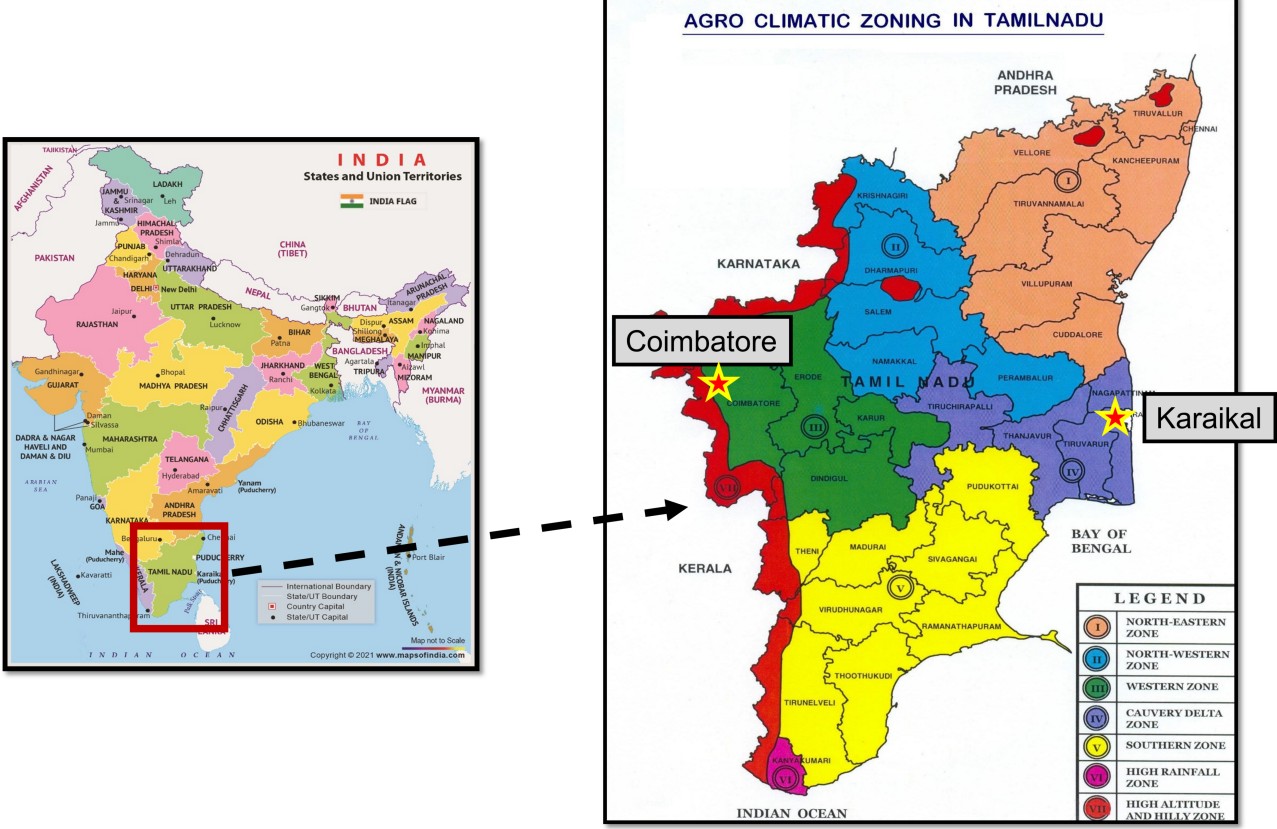

**Figure 1.** Location map of the study regions: Sadivayal, Coimbatore (western zone), and Karaikal (Cauvery Delta zone) in the Tamil Nadu state of India.

**Table 1.** Sowing and harvesting time of major rice-growing seasons in Tamil Nadu.

| District | Season | Sowing Time | Harvesting Time | Duration (Days) |
|---|---|---|---|---|
| Rainfed rice (Western zone) | Kar | May–Jun | Aug–Sep | <120 |
| | Samba | Aug | Dec–Jan | 130–135 and >150 |
| | Thaladi | Sep–Oct | Jan–Feb | 130–135 |
| | Navarai | Dec–Jan | March–Apr | <120 |
| Irrigated rice (Cauvery Delta zone) | Samba | Aug | Dec–Jan | 130–135 and >150 |
| | Late Thaladi | Oct–Nov | Jan–Feb | 115–120 |
| | Kuruvai | Jun–July | Sep–Oct | <120 |
| | Navarai | Dec–Jan | March–Apr | <120 |

### 2.2. Methodology

The following framework was adopted to evaluate rainfall contribution in rainfed and irrigated rice cultivation: (1) assessment of long-term rainfall variability in terms of annual, seasonal, monthly, and weekly time-scale; (2) investigation of rainfall trend;

(3) determination of sowing and harvesting time based on the onset and withdrawal of monsoon; (4) identifying the frequency of dry- and wet-spell occurrence during monsoon; (5) finding the occurrence of assured rainfall magnitude at different probability levels using incomplete gamma distribution; and (6) estimating water availability and shortage at different growth stages of rice through rainfall classification analysis (Figure 2). Daily rainfall data were collected for both locations covering the period from 1951 to 2015.

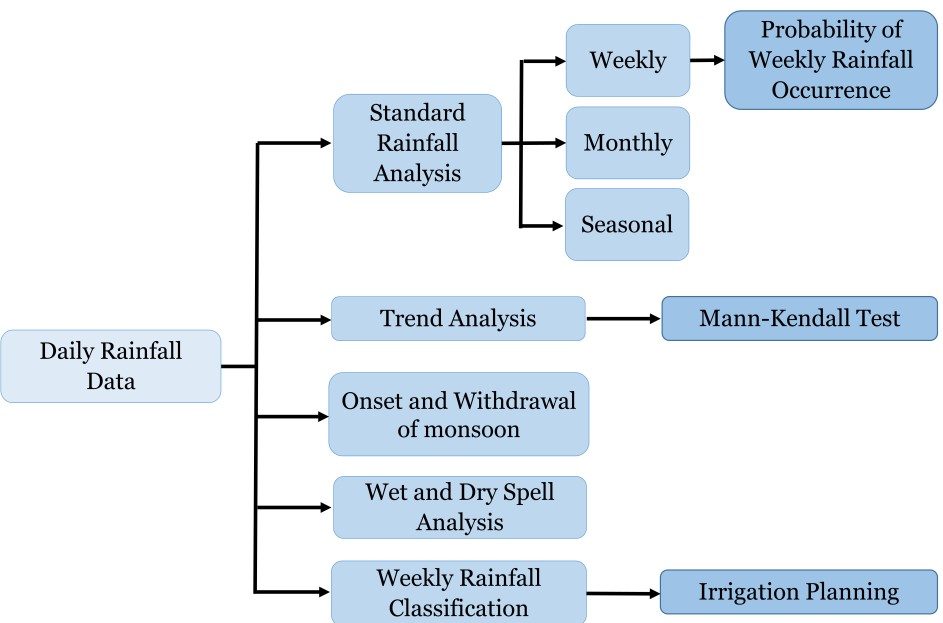

**Figure 2.** Schematic illustration of the rainfall analysis implemented in this study.

### 2.2.1. Standard Rainfall Analysis

The long-term rainfall variability was assessed on seasonal, monthly, and weekly time-scale to understand the rainfall distribution to rice growth during different phenological stages. Annual and seasonal rainfall variability is categorized into three degrees of variability based on the coefficient variation, as follows: low variability for CV < 20, moderate variability for 20 < CV < 30, and high variability for CV > 30 [32]. Instead of standard crop growing seasons, i.e., Kharif (June–October) and Rabi (October–February), our analysis focused on the major local rice-growing seasons, namely, Kar, Thaladi, Kuruvai, and Late Thaladi (Table 1). Data from daily rainfall are converted to weekly rainfall (standard meteorological week (SMW)) data, indicating weekly rainfall availability at various crop growth stages.

### 2.2.2. Rainfall Trend Analysis

The *Mann–Kendall (MK)* test was performed to analyze statistical significance for all annual, seasonal, and monthly rainfall trends, together with Sen's slope estimator for 65 years of time-series data (1951–2015). Mann–Kendall was employed to assess the trend, and the Z score was used to determine positive or negative trends. If the Z statistic is positive, the trend of the data series is considered ascending, and the trend is considered declining if the Z statistic is negative.

*Sen's Slope* (Q) is the robust estimate of the trend magnitude. A positive value of the median of Q shows the ascending trend, and a negative value shows the descending trend. Confidence limits and Sen's slope are the two important parameters in this test. The true values of slope (change per unit time) were predicted using Sen's slope estimator in mm year$^{-1}$; a positive (negative) value determines an increased value (decreased value) in the trend. Slope values are acceptable and reliable in the region of the confidence limits (1 and 5%).

### 2.2.3. Onset and Withdrawal of Monsoon

The onset and cessation of the rainy season were analyzed using the forward and backward accumulation method [33]. The date of rainfall onset was determined by the forward accumulation of weekly rainfall until 75 mm rainfall accumulated, starting from the normal onset date, on which the locals have been practicing. This volume of rain (75 mm) has been intended for the onset time of sowing rainfed rice. The date of rainfall recession was based on a rainfall accumulation of 20 mm summed backward daily.

### 2.2.4. Wet- and Dry-Spell Analysis

A day was considered a "dry day" when daily rainfall was less than 0.1 mm, and successive four continuous dry days were counted as a dry spell (DS). The reverse condition was contemplated as a wet day (rainfall $\geq$ 0.1 mm) and a wet spell (four successive wet days). Alternate wetting and drying pattern of irrigation is majorly followed in Tamil Nadu. The rice field is flooded once in three (dry weather) to five days (humid conditions). The 3-, 5-, and 7-day dry spells along with moderate and extreme wet spells were determined in different rice-growing stages. Further, dry periods with distinct lengths of 5, 6–10, 11–15, 16–20 days, and more than 20 days were calculated to better understand the potential risk of water stress during the rice-growing seasons.

### 2.2.5. Probability Distribution for Weekly Rainfall

The incomplete gamma probability distribution is used to determine the occurrence of rainfall events at different probability levels and calculate the percentage probability of occurrence of desired weekly rainfall amount for crop planning. The weekly rainfall data were analyzed at 10, 25, 50, 75, and 90 percent probability levels.

### 2.2.6. Irrigation Planning

The following thresholds were considered for the rainfall classification to plan the irrigation schedule: (i) water stress (<5 mm, severely dry), (ii) avoid stress (5–25 mm, slightly dry), (iii) sufficient (25–50 mm), and (iv) excess (>50 mm). These thresholds were categorized based on rice water demand and available rainfall during the particular week.

## 3. Results

### 3.1. Standard Rainfall Analysis

The standard rainfall analysis assessed the average rainfall distribution at four different time scales: annual, seasonal, monthly, and weekly. The comparison of mean annual, seasonal, and monthly rainfall for the period 1951 to 2015 is shown in Figure 3. Annual rainfall from 1951 to 2015 in Sadivayal varied in the range from a maximum of 2002.4 mm in 1971 to a minimum of 280.4 mm in 1974, with a mean of 721.9 mm and a coefficient of variation (CV) of 36%. On the other hand, the mean annual rainfall in the Karaikal region was 1108.5 mm in the same period. The Karaikal annual rainfall varied from 600 mm in 1980 to 1752.4 mm in 2004.

According to the standard rainfall analysis, annual rainfall in the rainfed rice-growing region (Sadivayal) showed comparably higher CV (35.16%) than in the irrigated rice-growing region (Karaikal) (CV = 25.14%), implying more interannual variation in annual rainfall. On the other hand, the values of average annual and seasonal rainfall are higher in the irrigated rice region.

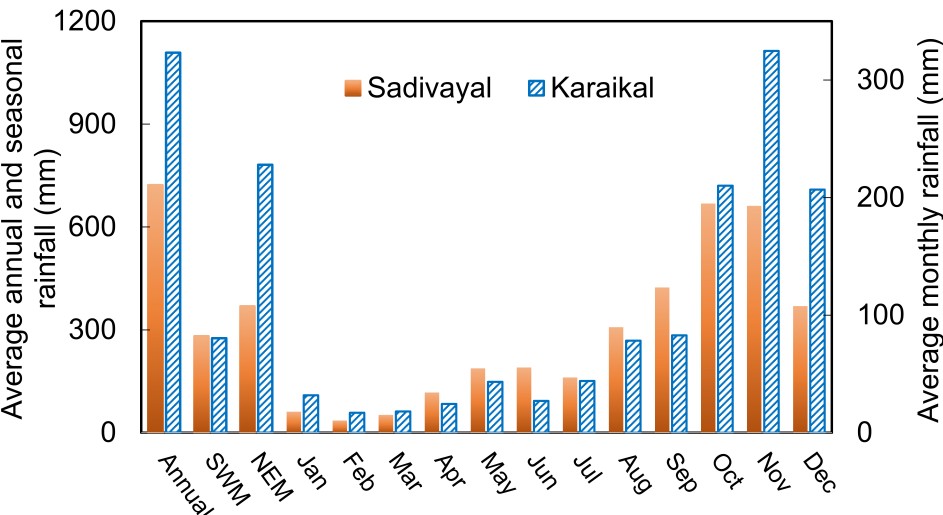

**Figure 3.** Comparison of annual, seasonal, and monthly rainfall in Sadivayal and Karaikal.

### 3.1.1. Seasonal Rainfall Analysis

The mean rainfall received during the Kar season was 281.43 mm, and 368.73 mm in the Thaladi season in the Sadivayal region. In the Karaikal region, the mean seasonal rain in the Kuruvai and Late Thaladi seasons was 275.56 and 781.46 mm, respectively. The inter-seasonal variability was observed in the range of 0–50% deviations, except during extreme events (Figure 4). The fluctuations in rainfall in both locations vary owing to the region's topography and monsoon rainfall dynamics.

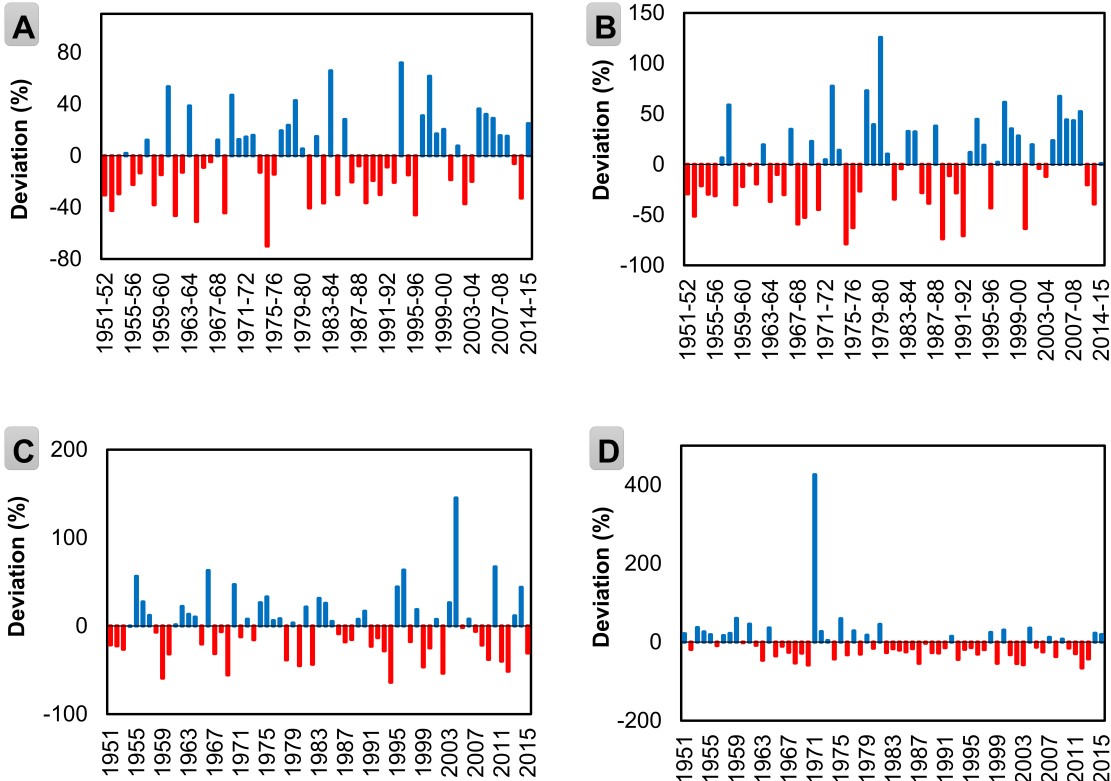

**Figure 4.** Seasonal rainfall deviation with the long-term mean of 65 years. (**A**) Late Thaladi Season (NEM) in Karaikal, (**B**) Thaladi Season (NEM) in Sadivayal, (**C**) Kuruvai Season (SWM) in Karaikal, and (**D**) Kar Season (SWM) in Sadivayal.

### 3.1.2. Monthly Rainfall Analysis

The maximum rainfall occurred in September (1385.30 mm), and October was the month with the highest percentage of rainfall contribution (25%) in the Sadivayal region (Figure 3). Similarly, October, November, and December have received the largest share of annual rainfall (more than 60%), and the highest percentage of rainfall contribution (30% annual contribution) occurred in November in the Karaikal region [34]. On the contrary, January (1%) and February (1.57%) months received insignificant rainfall for both irrigated and rainfed rice cultivation.

### 3.1.3. Weekly Rainfall Analysis

Standard weekly rainfall analysis of 65 years indicates that, with the exception of the Late Thaladi season (Karaikal), three other studied rice-growing seasons (i.e., Kar, Kuruvai, and Thaladi) require supplementary irrigation, which should be planned to compensate for water shortages during growth stages. The Late Thaladi season receives sufficient rainfall for rice growth (>50 mm) in most of the growing weeks (more than 60%), of which 45th SMW is the wettest week with the highest average weekly rainfall (93.2 mm). On the other hand, the final weeks of the Thaladi season receive minimal rain (51st until the 4th SMW), so rice crops are exposed to water stress at the ripening stage, and even worse, they may suffer from extensively dry conditions during the reproductive phase of its growth when seedlings are mistakenly transplanted late. To avoid water stress during the Kuruvai season, supplementary irrigation is required in almost all weeks of the rice-growing season due to a lack of sufficient rainfall that causes deficits in soil moisture (Figure 5). The instability of weekly rainfall distribution (CV > 150%) in both the Kar and Kuruvai seasons leads to a rapid disappearance of scattered rainfall and causes a sporadic dry period. The standard weekly rainfall analysis also highlighted that the mean length of rainy seasons is 13 weeks (90 days), which implies a short-duration rice variety is preferable over medium and long-duration for this region.

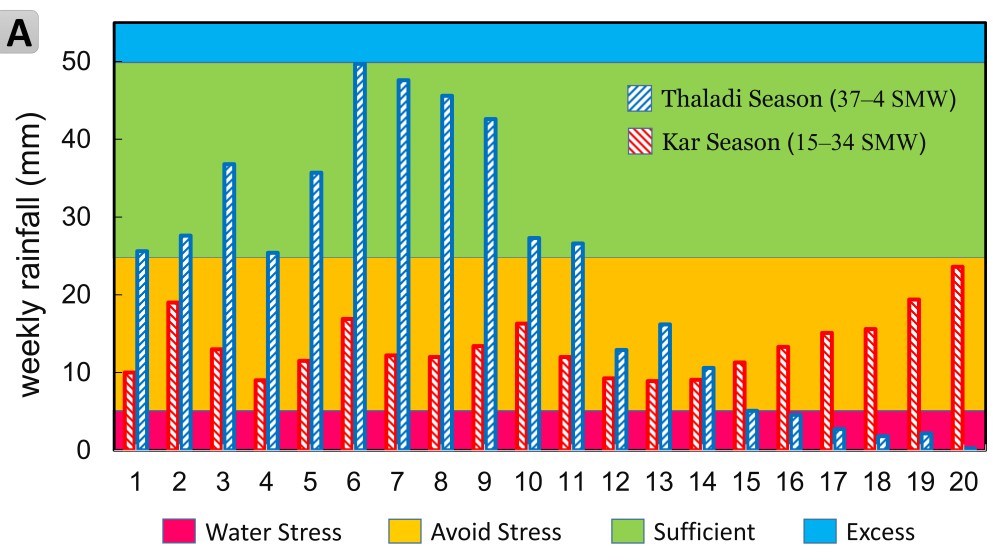

**Figure 5.** *Cont.*

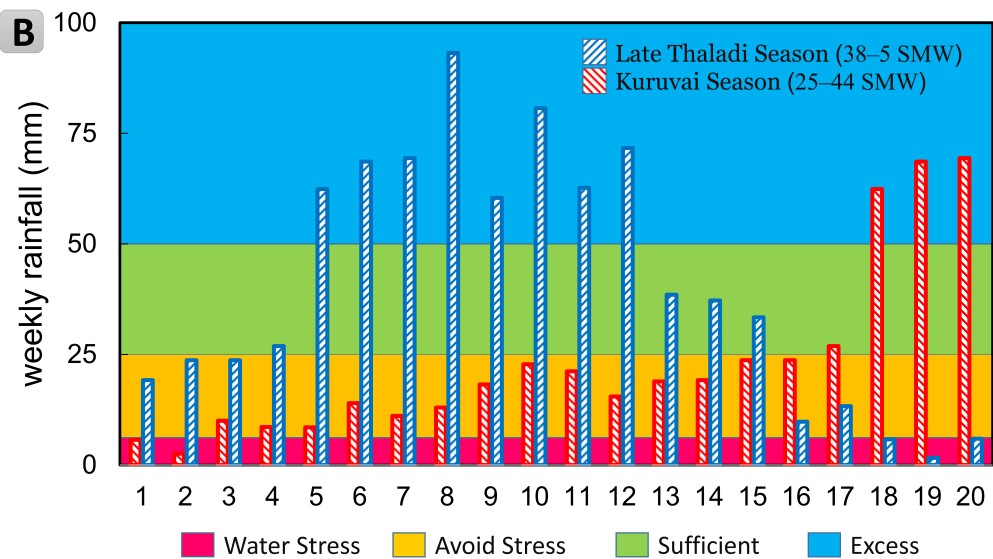

**Figure 5.** Average weekly rainfall distribution during major rice-growing seasons in (**A**) Sadivayal and (**B**) Karaikal.

*3.2. Rainfall Trend Analysis*

The time-series trend analysis over the rainfed and irrigated regions indicates that the annual rainfall trend based on the Mann–Kendall test (the Z value) is −2.12 for Sadivayal and 2.44 for the Karaikal region. The overall results of the Mann–Kendall test and statistical analysis at a seasonal time scale are shown in Tables 2 and 3, respectively. Seasonal rainfall in the Kar season has the sharpest decreasing trend (Z = −2.31), while the Late Thaladi season (Z = 2.36) shows the most positive trend. Long-term monthly trend analysis from 1951 to 2015 indicated that May, June, September, and October in both regions and November and December in the Karaikal region had shown an increasing trend. In both locations, a downward trend is observed during the southwest monsoon (SWM) months, i.e., June, July, and August [35]. Compared to the Kuruvai season, the Kar season has shown a more significant reduction in seasonal rainfall (diminishing with a magnitude of 1.52 mm per season). Thus, water stress is more likely to occur during SWM in the Sadivayal region than in the Karaikal region. The other critical rainfall indicators, such as rainy days, wet days, and wet spells that play a vital contribution to the rice water requirement and irrigation demand estimation, have shown a downward trend in both regions. On the other hand, the variables contributing to water stress analysis have shown an upward trend, i.e., dry days and dry spells (Table 2).

Rainfall in Sadivayal (during the Kar season), with a maximum of 52 rainy days and 138 wet days, is more intense than rainfall intensity in Karaikal (during the Kuruvai season), with a maximum of 35 rainy days. The downward trend (negative Z value) in wet-spell analysis during the study period is consistent with the decreasing number of rainy days in both the monsoon seasons. Therefore, rainfed rice was more affected due to changing pattern of rainfall. During the southwest monsoon, a decreasing rainfall trend is observed in Karaikal in the month of July with a magnitude of 0.5 mm month$^{-1}$. In Sadivayal, the volume decreased to 0.7, 0.5, and 0.6 mm month$^{-1}$, during June, July, and August, respectively. Showing a positive slope value, the trend of NEM was upward in both regions. Rainfall changes during NEM were found to be stable and reliable for both regions [36]. Overall, the monsoonal rainfall has shown an increasing trend during the winter season in both regions. Conversely, trend analysis (of the summer season indicates that the monsoon rainfall has weakened (less rainfall during May–September), and the distribution of precipitation within the monsoon season has become highly erratic.



**Table 2.** Rainfall pattern during major rice-growing seasons in Sadivayal and Karaikal.

| Rainfed Rice (Sadivayal) | Kar Season | | | | | | | Thaladi Season | | | | | | |
|---|---|---|---|---|---|---|---|---|---|---|---|---|---|---|
| | Ave. | Max. | Min. | SD | CV % | Z | Q | Ave. | Max. | Min. | SD | CV % | Z | Q |
| Seasonal rainfall (mm) | 281.43 | 1512.9 | 98.6 | 179.45 | 63.7 | −2.31 | −1.52 | 368.7 | 832.4 | 78.4 | 158.2 | 42.9 | 2.1 | 2.27 |
| Rainy day [a] | 23.05 | 52 | 10 | 8.25 | 35.8 | −2.55 | −0.14 | 21.27 | 42 | 7 | 6.77 | 31.84 | −3.38 ** | −0.23 |
| Wet day [a] | 56.06 | 138 | 20 | 32.27 | 57.5 | −3.99 ** | −0.58 | 34.53 | 62 | 12 | 11.8 | 34.16 | −4.59 ** | −0.63 |
| Wet spell | 21.38 | 112 | 0 | 32.55 | 152.2 | −3.66 ** | −0.24 | 11.14 | 37 | 0 | 9.14 | 82 | −4.39 ** | −0.39 |
| Dry day [a] | 96.94 | 133 | 15 | 32.27 | 33.29 | 3.99 ** | 0.58 | 116.72 | 139 | 89 | 11.81 | 10.11 | 4.64 | 0.63 |
| Dry spell | 54 | 98 | 0 | 24.13 | 44.68 | 3.99 ** | 0.64 | 87.3 | 117 | 55 | 13.43 | 15.38 | 5.06 ** | 0.69 |
| **Irrigated Rice (Karaikal)** | **Kuruvai Season** | | | | | | | **Late Thaladi Season** | | | | | | |
| | Ave. | Max. | Min. | SD | CV % | Z | Q | Ave. | Max. | Min. | SD | CV % | Z | Q |
| Seasonal rainfall (mm) | 275.56 | 681.8 | 100.7 | 101.85 | 36.96 | −0.39 | −0.28 | 781.46 | 1343.6 | 234.9 | 253.97 | 32.5 | 2.36 | 4.15 |
| Rainy day [a] | 22.62 | 35 | 10 | 6.76 | 29.87 | −1.83 | −0.09 | 37.09 | 58 | 18 | 9.15 | 24.66 | 0.3 | 0 |
| Wet day [a] | 56.51 | 83 | 10 | 19.33 | 34.2 | −3.11 * | −0.4 | 61.84 | 93 | 26 | 13.84 | 22.38 | −3.17 * | −0.25 [a] |
| Wet spell | 14.89 | 32 | 0 | 9.35 | 62.81 | −2.94 * | −0.21 | 31.56 | 65 | 5 | 12.81 | 40.59 | −1.9 | −0.18 |
| Dry day [a] | 96.49 | 143 | 70 | 19.33 | 20.03 | 3.11 * | 0.4 | 89.41 | 126 | 58 | 13.84 | 15.48 | 3.14 * | 0.25 |
| Dry spell | 48.45 | 125 | 11 | 26.34 | 54.37 | 2.75 * | 0.48 | 56.73 | 97 | 18 | 15.18 | 26.76 | 3.37 ** | 0.33 |

[a] number of days, * significant at 5%, ** significant at 1%.

### 3.3. Onset and Withdrawal of Monsoon

The onset of the monsoon is the key factor in determining the sowing date of the crop. Table 3 presents the early, normal, and delayed monsoon onset and withdrawal dates. According to the forward and backward accumulation method, the Late Thaladi season in the Karaikal starts from the 42nd SMW (October 15–21) and remains active up to the 2nd SMW (January 8–14). Considering the time interval between the early onset date and the late withdrawal date during the Late Thaladi season, the lead time is 119 days, which requires cultivation of short-duration rice varieties in this region.

The average onset date of the Thaladi season, which is the main rainy season for rainfed-rice cropping in Sadivayal, is October 1, with a standard deviation of 12 days [37]. Therefore, the earliest and the most delayed week of the onset date of the Thaladi season are the 38th week (September 17–23) and 42nd week (October 15–21), respectively. However, the withdrawal of monsoonal rain during the end of NEM varies in intensity and frequency (Table 2). Therefore, early sowing will avoid crop failure owing to the withdrawal of NEM with 23 days deviation from the normally expected termination date.

**Table 3.** Normal, early, and delayed onset and withdrawal of monsoon during major rice-growing seasons in Sadivayal and Karaikal.

| Rainfed Rice (Sadivayal) | | | Normal | Early | Delayed | SD |
|---|---|---|---|---|---|---|
| Kar | Onset | Day | 8-Jun | 10-May | 7-Jul | 29 |
| | | SMW | 23 | 20 | 28 | 4 |
| | Withdrawal | Day | 10-Sep | 26-Aug | 24-Sep | 15 |
| | | SMW | 37 | 35 | 39 | 2 |
| Thaladi | Onset | Day | 1-Oct | 19-Sep | 14-Oct | 12 |
| | | SMW | 40 | 38 | 42 | 2 |
| | Withdrawal | Day | 23-Dec | 19-Nov | 25-Jan | 34 |
| | | SMW | 51 | 47 | 4 | 5 |
| **Irrigated Rice (Karaikal)** | | | **Normal** | **Early** | **Delayed** | **SD** |
| Kuruvai | Onset | Day | 2-Aug | 13-Jul | 22-Aug | 20 |
| | | SMW | 31 | 29 | 34 | 3 |
| | Withdrawal | Day | 26-Oct | 22-Oct | 31-Oct | 4 |
| | | SMW | 44 | 43 | 44 | 1 |
| Late Thaladi | Onset | Day | 16-Oct | 8-Oct | 25-Oct | 8 |
| | | SMW | 42 | 41 | 43 | 1 |
| | Withdrawal | Day | 10-Jan | 18-Dec | 1-Feb | 23 |
| | | SMW | 2 | 51 | 5 | 3 |

### 3.4. Wet- and Dry-Spell Analysis

The results of the Mann–Kendal test depict that wet spells with different lengths of 3, 5, and 7 days have a negative long-term trend in the four cropping seasons, in which the Kar season has experienced the most declining slope among them. The decreasing trend of 3-, 5-, and 7-day wet spells (negative Z values) in the Karaikal is slower than the negative trend of the wet spells in the Sadivayal (Table 4). Given the importance of short dry spells in adopting sustainable strategies, the increasing slope of the 3-day dry spell is less than the 5-day dry spells in all the growing seasons, except the Late Thaladi season showing more increasing trend of shortest dry spell. A dry period of one week with the fastest uptrend (Z = 4.15) has been observed in the Kar season, indicating that the longest dry spell (7 days) has become more widespread in this growing season.

**Table 4.** Characteristics of dry and wet spells during major rice-growing seasons in Sadivayal and Karaikal.

| Rainfed Rice (Sadivayal) | Kar Season | | Thaladi Season | |
|---|---|---|---|---|
| | **Test Z** | **Q** | **Test Z** | **Q** |
| **3 WS** | −4.00 *** | −0.31 | −2.53 * | −0.20 |
| **5 WS** | −3.62 *** | −0.16 | −2.42 * | −0.12 |
| **7 WS** | −4.05 *** | −0.10 | −2.34 * | −0.30 |
| **3 DS** | 3.60 *** | 0.56 | 2.90 ** | 0.30 |
| **5 DS** | 3.75 *** | 0.56 | 3.21 ** | 0.30 |
| **7 DS** | 4.15 *** | 0.50 | 3.33 *** | 0.32 |
| **Irrigated Rice (Karaikal)** | **Kuruvai Season** | | **Late Thaladi Season** | |
| | **Test Z** | **Q** | **Test Z** | **Q** |
| **3 WS** | −3.01 ** | −0.3 | −2.21 * | −0.22 |
| **5 WS** | −2.85 ** | −0.15 | −1.80 + | −0.16 |
| **7 WS** | −3.07 ** | −0.07 | −1.69 + | −0.13 |
| **3 DS** | 2.74 ** | 0.48 | 3.53 *** | 0.31 |
| **5 DS** | 2.81 ** | 0.47 | 3.26 ** | 0.33 |
| **7 DS** | 2.68 ** | 0.39 | 3.17 ** | 0.31 |

+ significance at 10% level, * significance at 5% level, ** significance at 1%, *** significance at 0.1%.

The total number of occurrences of dry spells with varying lengths of less than 6, 6–10, 11–15, 16–20, and more than 20 days across the years was calculated to determine the probable water stress during the rice growth stages (Table 5). The dry-spell analysis in the irrigated rice farming region reveals an increasing number of days without rain, though the total annual rainfall shows a rising trend. This capricious rainfall pattern triggers extreme climatic events such as floods and droughts. Precipitation distribution in the western zone is more erratic than rainfall distribution in Karaikal, mainly in the earlier months of the NEM seasons. On the other hand, SWM rainfall distribution was more consistent, and intermittent dry spells occurred evenly in both study areas. Further, the inconsistent long-lead dry spells (more than 20 days) occurring during SWM are observed in the Karaikal.

**Table 5.** Prolonged dry-spell category analysis during major rice-growing seasons in Sadivayal and Karaikal.

| | Dry Days in Sadivayal | | | | | Dry Days in Karaikal | | | | |
|---|---|---|---|---|---|---|---|---|---|---|
| **NEM Season** | **<5** | **6–10** | **11–15** | **16–20** | **>20** | **<5** | **6–10** | **11–15** | **16–20** | **>20** |
| October | 29 | 3 | 1 | 0 | 0 | 31 | 1 | 1 | 0 | 0 |
| November | 25 | 5 | 2 | 1 | 0 | 30 | 2 | 0 | 0 | 0 |
| December | 17 | 7 | 4 | 2 | 3 | 28 | 4 | 1 | 0 | 0 |
| January | 6 | 4 | 5 | 4 | 15 | 18 | 7 | 4 | 2 | 3 |
| February | 5 | 2 | 2 | 2 | 19 | 9 | 5 | 4 | 3 | 10 |
| **SWM Season** | **<5** | **6–10** | **11–15** | **16–20** | **>20** | **<5** | **6–10** | **11–15** | **16–20** | **>20** |
| May | 24 | 6 | 2 | 1 | 0 | 21 | 5 | 3 | 1 | 4 |
| June | 24 | 5 | 2 | 1 | 0 | 23 | 5 | 2 | 1 | 2 |
| July | 27 | 4 | 2 | 1 | 0 | 26 | 3 | 1 | 1 | 2 |
| August | 26 | 5 | 2 | 1 | 0 | 29 | 3 | 1 | 0 | 1 |
| September | 23 | 5 | 3 | 1 | 1 | 29 | 3 | 1 | 0 | 0 |

The Kuruvai season has experienced the maximum number of long-term dry spells (more than 20 days) at the start of SWM. It is quite evident that the latter part of the SWM season experiences lesser long-term dry spells compared to NEM. The driest months during the NEM season are January and February. Thus, the long-term dry spells during these months most often occur in the Sadivayal region.

### 3.5. Incomplete Gamma Probability

Weekly rainfall of more than 50 mm is observed during 16th, 18th, and from 38th to 47th SMW, and also from 41st to 45th SMW with 10% and 25% probability in the western zone, respectively (Figure 6A,B), while the earliest weekly rainfall of more than 25 mm to less than 50 mm (sufficient water for rice crop) is observed during the 16th and 8th SMW at 10% and 25% probability confidence levels, respectively. On the other hand, from 34th to 52nd SMW, more than 50 mm rainfall is available at a 10% probability confidence level for irrigated rice cultivation. Therefore, compared to the rainfed rice cultivation in Sadivayal, the irrigated rice cultivation period is more extended in Karaikal (Figure 6C,D). Thus, the medium duration (121–140 days) and long-duration (141–160 days) rice varieties are suitable for irrigated rice cultivation. On the other hand, rainfall within the range of 25 to 50 mm is considered an adequate weekly rainfall for rice growth recorded from 18th SMW onwards and contributed up to 38th SMW of the Late Thaladi season in the Cauvery Delta zone.

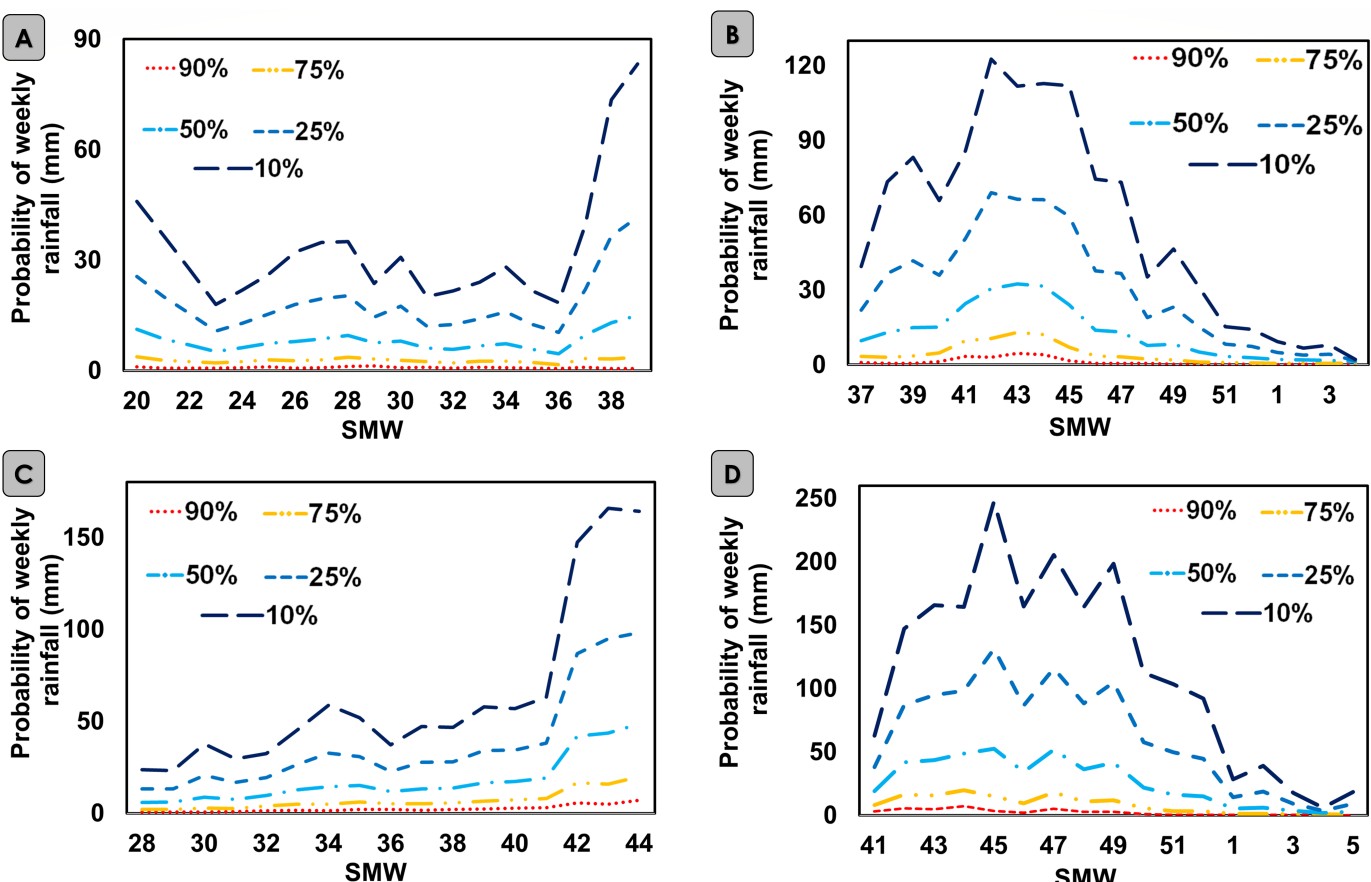

**Figure 6.** Weekly assured rainfall at different probability levels. (**A**) Kar season (Sadivayal), (**B**) Thaladi season (Sadivayal), (**C**) Kuruvai season (Karaikal), and (**D**) Late Thaladi season (Karaikal).

Excessive weekly rainfall (more than 50 mm) with a probability level of 25 percent occurs in the 41st and 42nd SMW during the Thaladi and Late Thaladi seasons. Incomplete gamma probability analysis of annual rainfall indicates that water requirement supported by rain for the irrigated Kuruvai and Late Thaladi rice (between 700 to 1100 mm) is available at more than 50 percent probability. Nevertheless, the same amount of water is available at less than 25 percent probability for the Kar and Thaladi rice-growing seasons in the Sadivayal region.

*3.6. Rice-Growth Stagewise Irrigation Planning Based on the Rainfall Classification Analysis*

The vegetative phase (sowing to panicle initiation (PI)), reproductive phase (PI to flowering), and ripening phase (flowering to harvesting) are the three major growth stages analyzed in this study. Further, four categories of water availability were considered with the classification of weekly rainfall. Table 6 shows the distribution frequency of weekly rainfall, categorized into water stress, avoid stress, sufficient, and excess water availability in each rice growth stage. Water stress condition (weeks with less than 5 mm rainfall) is prevalent with a high frequency in most of the weeks of the Kar season. Moreover, from 26th to 28th SMW, sufficient water is rarely available when the rice crop is sensitive to water stress. Similarly, severe water stress occurs frequently during the first weeks of the Kuruvai season from 28th to 30th SMW (Table 6A,C). Despite the frequent water stresses observed during the weeks of the Kar and Kuruvai seasons, the number of events with sufficient rainwater is significant enough to be considered to meet the rice water demand. On the other hand, most of the weeks of the Thaladi and Late Thaladi seasons were reliable, with a high frequency of more than 50 mm water availability during 40 years. However, stressful conditions due to the water shortage are expected in the last weeks of the Thaladi and Late Thaladi seasons (Table 6B,D). The late sowing rice is exposed to definite water stress during its critical growth stages.

**Table 6.** Water availability assessment based on weekly rainfall classification.

**(A) Kar season (Sadivayal)**

| | | 20 | 21 | 22 | 23 | 24 | 25 | 26 | 27 | 28 | 29 | 30 | 31 | 32 | 33 | 34 | 35 | 36 | 37 | 38 | 39 |
|---|---|---|---|---|---|---|---|---|---|---|---|---|---|---|---|---|---|---|---|---|---|
| Classification | Normal | | | | | | | | VS | | | | | | RES | | | | RIS | | |
| | Early | | | | VS | | | | | | RES | | | | RIS | | | | | | |
| | Late sowing | | | | | | | | | | | VS | | | | RES | | | | RIS | |
| **WR (mm)** | **Kar SMW** | **20** | **21** | **22** | **23** | **24** | **25** | **26** | **27** | **28** | **29** | **30** | **31** | **32** | **33** | **34** | **35** | **36** | **37** | **38** | **39** |
| <5 | Water stress | 20 | 22 | 16 | 17 | 21 | 17 | 26 | 26 | 28 | 22 | 23 | 19 | 23 | 15 | 17 | 16 | 16 | 15 | 15 | 13 |
| 5–25 | Avoid stress | 7 | 10 | 16 | 13 | 10 | 16 | 8 | 10 | 8 | 12 | 12 | 12 | 12 | 14 | 12 | 16 | 11 | 13 | 10 | 7 |
| 25–50 | Sufficient | 11 | 4 | 5 | 7 | 5 | 3 | 4 | 3 | 3 | 3 | 4 | 4 | 3 | 8 | 6 | 1 | 5 | 5 | 5 | 5 |
| >50 | Excess | 2 | 4 | 3 | 3 | 4 | 4 | 2 | 1 | 1 | 3 | 1 | 5 | 2 | 3 | 5 | 7 | 8 | 7 | 10 | 15 |

**(B) Thaladi season (Sadivayal)**

| | | 37 | 38 | 39 | 40 | 41 | 42 | 43 | 44 | 45 | 46 | 47 | 48 | 49 | 50 | 51 | 52 | 1 | 2 | 3 | 4 |
|---|---|---|---|---|---|---|---|---|---|---|---|---|---|---|---|---|---|---|---|---|---|
| Classification | Normal | | | | | | VS | | | | RES | | | | RIS | | | | | | |
| | Early | | VS | | | | RES | | | | RIS | | | | | | | | | | |
| | Late sowing | | | | | | | | | | | VS | | | | RES | | | | RIS | |
| **WR (mm)** | **Thaladi SMW** | **37** | **38** | **39** | **40** | **41** | **42** | **43** | **44** | **45** | **46** | **47** | **48** | **49** | **50** | **51** | **52** | **1** | **2** | **3** | **4** |
| <5 | Water stress | 15 | 15 | 13 | 15 | 4 | 12 | 2 | 8 | 7 | 16 | 19 | 22 | 25 | 28 | 31 | 33 | 36 | 37 | 36 | 38 |
| 5–25 | Avoid stress | 13 | 10 | 7 | 7 | 15 | 6 | 11 | 8 | 5 | 11 | 5 | 8 | 12 | 8 | 7 | 4 | 1 | 1 | 1 | 1 |
| 25–50 | Sufficient | 5 | 5 | 5 | 12 | 7 | 5 | 10 | 8 | 10 | 4 | 7 | 8 | 0 | 0 | 0 | 1 | 1 | 1 | 1 | 0 |
| >50 | Excess | 7 | 10 | 15 | 5 | 13 | 16 | 16 | 15 | 17 | 8 | 8 | 1 | 2 | 3 | 1 | 1 | 1 | 0 | 1 | 0 |

**(C) Kuruvai season (Karaikal)**

| | | 28 | 29 | 30 | 31 | 32 | 33 | 34 | 35 | 36 | 37 | 38 | 39 | 40 | 41 | 42 | 43 | 44 |
|---|---|---|---|---|---|---|---|---|---|---|---|---|---|---|---|---|---|---|
| Classification | Normal | | | | | | VS | | | | RES | | | | RIS | | | |
| | Early | | | | VS | | | | RES | | | | RIS | | | | | |
| | Late sowing | | | | | | | | | VS | | | | RES | | | RIS | |
| **WR (mm)** | **Kuruvai SMW** | **28** | **29** | **30** | **31** | **32** | **33** | **34** | **35** | **36** | **37** | **38** | **39** | **40** | **41** | **42** | **43** | **44** |
| <5 | Water stress | 24 | 21 | 21 | 18 | 18 | 13 | 14 | 11 | 13 | 11 | 11 | 11 | 12 | 6 | 7 | 3 | 3 |
| 5–25 | Avoid stress | 13 | 13 | 15 | 17 | 16 | 17 | 13 | 19 | 19 | 17 | 16 | 13 | 12 | 15 | 5 | 7 | 8 |
| 25–50 | Sufficient | 2 | 6 | 0 | 3 | 5 | 7 | 8 | 7 | 7 | 11 | 9 | 8 | 13 | 10 | 14 | 10 | 6 |
| >50 | Excess | 1 | 0 | 4 | 2 | 1 | 3 | 5 | 3 | 1 | 1 | 4 | 8 | 2 | 8 | 13 | 19 | 22 |

**Table 6.** *Cont.*

| Classification | | | | | | | | | | | | | | | | | | |
|---|---|---|---|---|---|---|---|---|---|---|---|---|---|---|---|---|---|---|
| (**D**) Late Thaladi season (Karaikal) | | | | | | | | | | | | | | | | | | |
| | Normal | | | | VS | | | | RES | | | RIS | | | | | | |
| Classification | Early | VS | | | RES | | | | RIS | | | | | | | | | |
| | Late sowing | | | | | VS | | | | | | | RES | | | | RIS | |
| **WR (mm)** | **Late Thaladi SMW** | 41 | 42 | 43 | 44 | 45 | 46 | 47 | 48 | 49 | 50 | 51 | 52 | 1 | 2 | 3 | 4 | 5 |
| <5 | Water stress | 6 | 7 | 3 | 3 | 5 | 12 | 6 | 9 | 7 | 14 | 16 | 17 | 28 | 26 | 31 | 36 | 35 |
| 5–25 | Avoid stress | 15 | 5 | 7 | 8 | 4 | 7 | 6 | 5 | 8 | 7 | 7 | 12 | 8 | 9 | 4 | 3 | 3 |
| 25–50 | Sufficient | 10 | 14 | 10 | 6 | 6 | 3 | 4 | 6 | 10 | 8 | 6 | 4 | 1 | 2 | 4 | 0 | 0 |
| >50 | Excess | 8 | 13 | 19 | 22 | 24 | 17 | 23 | 19 | 14 | 10 | 10 | 6 | 2 | 2 | 0 | 0 | 1 |

VS: vegetation stage, RES: reproduction stage, RIS: ripening stage.

## 4. Discussion

Sustainable rice production strongly depends on the monsoon rainfall patterns in peninsular India. This study examined the rainfall variability to determine potential sustainable measures to maximize monsoon rain utilization and augment rainwater's contributive role to rainfed and irrigated rice farming through proper crop planning.

As rice yields are highly responsive to seasonal rainfall anomalies [38], the Karaikal region has shown minimal interannual and interseasonal variability compared to Sadivayal. This seasonal and interannual variation makes the Sadivayal rice farming region more vulnerable to climate change anomalies. Thus, the irrigated rice cultivated in Karaikal is less affected by seasonal anomalies than rainfed rice practiced in Sadivayal [39]. For instance, harvesting and storing the surplus rainwater during high rainfall years (i.e., 1960–1961, 1983–1984, and 1993–1994) can aid rice crops to survive during the subsequent unforeseen dry seasons (i.e., 1961–1962, 1984–1985, and 1994–1995).

In a similar scenario, the water harvesting strategy could be adopted for years experiencing wet Thaladi seasons (i.e., 1987–1988) to avoid water stress caused by arduous drought conditions during the Kar season in consecutive years (i.e., 1989). Frequent dry anomalies during the Kuruvai season in the Cauvery Delta region for more than two years may be accompanied by increased rice field expansion to compensate for lower yields [38].

Anand and Karunanidhi [40] found similar results, indicating that a significant portion of annual rainfall (more than 80%) is received during the NEM months, including October, with the highest range in Tamil Nadu. Therefore, NEM (the Thaladi and Late Thaladi seasons) plays a vital role in rice productivity, as it coincides with the most water-sensitive rice growth stages [36]. Erratic rainfall distribution and insufficient precipitation in the early weeks of the vegetative stage of the Kar season necessitates SI frequently during 25th to 30th SMWs. Similarly, the reproductive period of the maturity stage of rice growth is faced with minimal rainfall resulting in assured water stress during the 48th and 49th SMWs of the Thaladi season. It was also observed that change in intensity and dispensation of weekly precipitation causes interim dry spells within a typical month of the rainy season. Based on the standard weekly rainfall analysis, short-duration rice varieties are recommended to be cultivated during the local monsoon seasons in Tamil Nadu.

A positive trend (Z value of 2.44) in annual rainfall is noticed in Karaikal, whereas Sadivayal has shown a downward (Z value of −2.12) [41]. With the decreasing trend of monthly rainfall from March to April, SI can be applied to provide adequate soil moisture for land preparation before transplanting seedlings in the Kuruvai season. With a diminishing magnitude of 1.52 mm per season, adopting SI is inevitably important in the Kar season to meet the rice water requirements. Without a well-scheduled SI, water stress is more likely to occur during rice growth in the Kar season. However, the rising trend of monthly rainfall in April can satisfy the soil moisture needed for land preparation in the Kar season.

On the other hand, Rani et al. [35] noticed an increasing trend in seasonal (the Thaladi season) and annual rainfall in Coimbatore city at the confidence level of 95 percent. The overall trend analysis shows that the total rainfall volume will decrease with a lower frequency and higher intensity, which may affect a prolonged dry spell. Similarly, with prolonged dry spells, the area covered by Thaladi/Late Thaladi rice has decreased compared to the previous seasons [42]. As the trend of rainy and wet days in the months of Thaladi, Late Thaladi, and Kuruvai seasons tend to decrease, the probability of encountering sporadic dry spells would increase. Therefore, it is recommended to adopt water harvesting and deficit irrigation methods in wet spells to manage probable dry periods during the rice-growing season.

The wet- and dry-spell analysis highlights the importance of SI, mainly for rainfed rice cultivation. The long dry spells in the first month of SWM might affect the active vegetative stage of rice growth, and farmers consider an irrigation plan at this stage to combat any adverse effects of climatic events on rice yields. In a similar pattern, the prolonged dry spells (>20 days) in the Thaladi and Late Thaladi seasons (NEM) increase irrigation demand

during the flowering to maturity period. Utilizing the extra water harvested during a wet spell to meet rice crop water requirements in subsequent periods with insufficient rainfall is a preferred cropping strategy for rainfed and irrigated rice cultivation regions. In doing so, small water tank reservoirs or in-farm ponds are the best choices. The maturity stage of rice growth faces probable water stress during NEM since the intensity of rainfall decreases and rainfall distribution weakens in December and January. Farmers are recommended to adopt the scheduled early sowing method for land preparation and transplanting seedlings to avoid probable water stress during the January and February months of the Thaladi and Late Thaladi seasons. The frequencies of short- (<5 days) and mid-term (6–10 days) dry-spell occurrence make farmers more inclined to adopt alternate wet/dry irrigation during monsoon time. This is considered a water-saving system for rice irrigation that helps farmers overcome water scarcity during the severe dry spells in May and June in the Kar and Kuruvai seasons [43].

The incomplete gamma probability for weekly rainfall in the Sadivayal region indicates that the appropriate date for land preparation is 16th and 38th SMW due to receiving 25 mm rainfall with a probability of 25% in the Kar and Thaladi seasons, respectively. The assured minimum rainfall with a 50 percent probability during 41st and 42nd SMW is suggested as an ideal sowing period for the Thaladi and Late Thaladi rice-growing seasons [44]. Weekly rainfall classification analysis shows that water scarcity during the vegetative growth phase (from 26th to 29th SMW) can be addressed utilizing rainwater harvested from 24th and 25th SMW as life-saving irrigation for rainfed rice cultivated in the Sadivayal region. The results of the weekly rainfall classification of the earlier weeks in the Kar season show that the early sowing ensures all water-sensitive stages of rainfed rice with sufficient rainwater. However, early-sown rice suffers from water stress due to the minimal availability of weekly rainwater during the critical vegetative stage of rice growth (from 28th to 30th SMW) in the Kuruvai season.

In contrast to the Kar and Kuruvai seasons, excessive water is available in most of the weeks of the Thaladi and Late Thaladi seasons. However, in the case of late sowing during the Thaladi and Late Thaladi seasons, water stress is unavoidable during the rice crop's panicle initiation stage and maturity phase (coinciding with 2nd to 4th SMW). Late sowing of rice results in water stress, which severely affects the panicle initiation and leads to reduced yield. To overcome this issue, government agencies suggest location-specific drought-tolerant rice cultivars (e.g., Pichavari and Poongkar) to avoid the adverse effects of erratic distributed rainfall such as water stress that occurs during the growth stages of rice, which is a water-sensitive crop. Overall, this study highlighted that improving water management planning by incorporating the rainfall contribution during rice growth is essential for long-term sustainable productivity in both rainfed and irrigated rice cultivation regions.

## 5. Conclusions

This study proposed sustainable solutions by examining the intensity and distribution of weekly rainfall for two distinct rice-growing systems in peninsular India. The analysis of dry and wet spells, identification of the onset date of monsoon, probability of rainfall occurrence, and classification of weekly rainfall are the most critical elements in the crop planning framework through which sustainable rice cultivation can be achieved. For instance, rainfall analysis in the Sadivayal region showed an uneven distribution of rainfall during the Kar and Thaladi rice-growing seasons. Therefore, rainfed rice cultivated in Sadivayal suffers from the adverse effects of rainfall aberrations on rice crop growth. However, the Karaikal region displays minimum rainfall variability and sporadic water stress during the crop growth period. October and November consistently received higher rainfall compared to January and February.

Weekly rainfall classification showed that harvesting the excess rainwater during the 45th SMW (93.2 mm) of the Late Thaladi rice (Karaikal) mitigates the risk of water stress in the leg vegetative phase (46th SMW) of its growth that experiences frequent water stress. On the other hand, an adaptive strategy of early sowing (38th SMW) is proposed for the rainfed

Thaladi rice (Sadivayal) that helps to avoid terminal water stress during the flowering stage with a satisfying rainfall contribution in the earlier weeks. Therefore, the proposed approach can be effectively used for water management and irrigation planning practices to enhance the sustainability of rainfed and irrigated rice cultivation. This framework can be further implemented in other rice-growing regions in peninsular India.

**Author Contributions:** Conceptualization, V.S.M.; methodology, M.K.B., V.S.M. and S.M.; data collection, V.S.M., A.N., P.S. and S.M.; data analysis, M.K.B.; writing—original draft, M.K.B. and V.S.M.; writing—review and editing, M.K.B., V.S.M., M.R.N., P.S., A.N. and S.M.; supervision and monitoring, V.S.M., M.R.N. and S.M. All authors have read and agreed to the published version of the manuscript.

**Funding:** This project has been funded by the E4LIFE International Doctoral Fellowship Program offered by Amrita Vishwa Vidyapeetham.

**Data Availability Statement:** The corresponding author can provide the data used in this work upon request.

**Acknowledgments:** Masoud Barati is funded by the E4LIFE International Ph.D. program offered by Amrita Vishwa Vidyapeetham.

**Conflicts of Interest:** The authors declare no conflict of interest.

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
