# Peer review of "Rainfall Variability and Rice Sustainability: An Evaluation Study of Two Distinct Rice-Growing Ecosystems"

_land, doi:10.3390/land11081242_

Round 1

Reviewer 1 Report

Dear authors and editors

The quality of the paper is very high and should be published

I recommend to mention that the paper is within the nature based solutions that we are developing in agriculture around the world

see here an example of professor Keesstra

Keesstra, S., Nunes, J., Novara, A., Finger, D., Avelar, D., Kalantari, Z., & Cerdà, A. (2018). The superior effect of nature based solutions in land management for enhancing ecosystem services. Science of the Total Environment610, 997-1009.

and Also you must mention that what you found is also researched in other crops 

see here a couple of examples from the Mediterranean

Cerdà, A., Daliakopoulos, I. N., Terol, E., Novara, A., Fatahi, Y., Moradi, E., ... & Pulido, M. 2021. Long-term monitoring of soil bulk density and erosion rates in two Prunus Persica (L) plantations under flood irrigation and glyphosate herbicide treatment in La Ribera district, Spain. Journal of Environmental Management, 282, 111965.

Cerdà, A., Novara, A., & Moradi, E. 2021. Long-term non-sustainable soil erosion rates and soil compaction in drip-irrigated citrus plantation in Eastern Iberian Peninsula. Science of The Total Environment787, 147549.

Cerdà, A., Terol, E., & Daliakopoulos, I. N. 2021. Weed cover controls soil and water losses in rainfed olive groves in Sierra de Enguera, eastern Iberian Peninsula. Journal of Environmental Management, 290, 112516.

Author Response

Dear reviewer, 

Reviewer 2 Report

General comments:

The purpose of this study is to evaluate the variation and probability of rainfall in the rice growing season, and its findings have important guiding significance for guiding local agricultural production. Overall, the writing level of the manuscript is good. But abstracts and figures need to be revised and refined before publication.

Special comments:

1. In the abstract part of the manuscript, the results are too simple, and valuable results and conclusions need to be supplemented.

2. The information in Figure 1 is too cumbersome. Only show information that is valuable to this study.

3. In Table 1, the season column, information does not match the title (i.e., season).

4. In Figure 1, the information represented by the Horizontal axis is explained.

5. Table 6 does not require color marks.

Author Response

Dear reviewer, 

Reviewer 3 Report

General comments

Please check the MDPI format on references!!

Please revise the article based on the MDPI format!!!

Please answer my questions:

What is the objective of this study?

What is the novelty of this study?

Why did you use your last articles to cite?

Specific comments

Line 16-17 revise sentences.

Line 41-42 revise sentences.

Line 86-91 revise sentence.

Line 150 slope values are acceptable and reliable in the region of the confidence limits (1 and 5%).

Line 166 In accordance with the change to by

Line 218 In a similar way changes to Similarly

Line 272 this?!!

Line 291-293 revise sentence.

Line 437-440 revise sentence.

Line 491- 496 revise sentence.

Please check the MDPI format in references.

 Please revise the article base on the English Editor native.

Author Response

Dear reviewer, 

Round 2

Reviewer 2 Report

The author addresses my concerns and recommends acceptance for publication.

Reviewer 3 Report

please revise based on English Editor